# Assessing Post-Fire Effects on Soil Loss Combining Burn Severity and Advanced Erosion Modeling in Malesina, Central Greece

**Ioanna Tselka [1,2,\*], Pavlos Krassakis [1,3], Alkiviadis Rentzelos [4], Nikolaos Koukouzas [3] and Issaak Parcharidis [1]**

[1] Department of Geography, Harokopio University of Athens, El. Venizelou 70, 17671 Athens, Greece; pkrassakis@hua.gr (P.K.); parchar@hua.gr (I.P.)
[2] School of Rural and Surveying Engineering, Zografou Campus, National Technical University of Athens, Iroon Polytechniou 9, 15780 Athens, Greece
[3] Centre for Research & Technology Hellas (CERTH), 15125 Athens, Greece; koukouzas@certh.gr
[4] Faculty of Geoscience, Utrecht University, Heidelberglaan 8, 3584 CS Utrecht, The Netherlands; a.rentzelos@students.uu.nl
\* Correspondence: ioannatselka@gmail.com

**Abstract:** Earth's ecosystems are extremely valuable to humanity, playing a key role ecologically, economically, and socially. Wildfires constitute a significant threat to the environment, especially in vulnerable ecosystems, such as those that are commonly found in the Mediterranean. Due to their strong impact on the environment, they provide a crucial factor in managing ecosystems behavior, causing dramatic modifications to land surface processes dynamics leading to land degradation. The soil erosion phenomenon downgrades soil quality in ecosystems and reduces land productivity. Thus, it is imperative to implement advanced erosion prediction models to assess fire effects on soil characteristics. This study focuses on examining the wildfire case that burned 30 km² in Malesina of Central Greece in 2014. The added value of remote sensing today, such as the high accuracy of satellite data, has contributed to visualizing the burned area concerning the severity of the event. Additional data from local weather stations were used to quantify soil loss on a seasonal basis using RUSLE modeling before and after the wildfire. Results of this study revealed that there is a remarkable variety of high soil loss values, especially in winter periods. More particularly, there was a 30% soil loss rise one year after the wildfire, while five years after the event, an almost double reduction was observed. In specific areas with high soil erosion values, infrastructure works were carried out validating the applied methodology. The approach adopted in this study underlines the significance of using remote sensing and geoinformation techniques to assess the post-fire effects of identifying vulnerable areas based on soil erosion parameters on a local scale.

**Keywords:** post-wildfire soil erosion; RUSLE (revised universal soil loss equation); burn severity; *NBR* (normalized burn ratio); earth observation; Copernicus images; GIS (geographic information systems); Central Greece

## 1. Introduction

Soil erosion constitutes one of the greatest global environmental threats. It occurs in the form of sheet, rill, and gully erosion and downgrades soil condition, water quality, species habitats, and the provision of ecosystem services [1]. A decrease in soil productivity threatens not only the balance but also the security of food production, which could lead to further financial consequences and might have a negative effect on people's life [2]. The main causes of soil erosion, such as tillage and rill erosion, are caused due to rainfalls and wind [3]. However, the study of wildfires resulting in soil erosion phenomena increasingly attracts scientific interest due to their adverse effects on soil [4]. Furthermore, in some particular ecosystems, wildfires are suggested to be the single most crucial cause of land

change [5]. Apart from the risk to lives and existing infrastructure, wildfires cause land and ecosystem degradation [6].

According to De Santis and Chuvieco [7], burn severity constitutes a key factor in evaluating post-fire incidents based on remote sensing data. Specifically, burn severity provides the basis in identifying fire effects in ecosystems with emphasis on the procedure of the examined ecosystem recovery. In this study, burn severity analysis demonstrates the evaluation of the burned area regarding the fire effects on the environment [8]. The energy generation due to the fuels flaming and combustion causes alterations in soil properties during a fire incident [5]. Vegetation and forest litter draining by fire enhances the exposure of the underlying soil matter to erosion, causing substantial changes to the physical and chemical soil properties [4].

It is estimated that more than half percent of the major ecosystems that produce food and feed all over the world have already been degraded. Thus, it is doubtful whether it could cover the requisite demand of the next decades [2]. To quantify the impact of soil loss and to develop effective measures for land conservation, many soil erosion models have been employed in assessing the risk of soil loss. That has been achieved through the use of common physical parameters found to be important from observational experience or multivariable statistical analysis such as slope, precipitation, vegetation cover, and soil erodibility [9]. Among multiple different models developed, the Water Erosion Prediction Project (WEPP), the universal soil loss equation (USLE) [10], and its revised version (RUSLE) [11]. These techniques demonstrate the most widely used empirical models in soil erosion research [1,4,9] through the estimation of long-term average rates, based on geomorphological (topography, land cover) and climatological (rainfall) features [6].

Even though RUSLE applications usually estimate soil loss at annual timescales, there are factors such as vegetation growth and rainfalls that are temporally variable. Therefore, to obtain a higher accuracy of soil erosion vulnerability assessment, it is important to implement the survey at smaller temporal scales, such as examining it from the aspect of season [12]. For instance, soil erosion risk considerably increases when a season of heavy rainfall coincides with low vegetation cover [13,14].

Over the last 30 years, the Mediterranean region has been vulnerable to climate change, particularly due to its sensitivity to drought and rising temperatures. Areas that belong to that region, such as Greece, Southern Italy, Southern France, and Spain, are facing great ecological disasters every year, especially due to wildfires and intense heatwaves. Under the prism of the climate crisis, the identification of the spatial impact of these wildfires is crucial operationally to identify vulnerable areas related to soil loss for different timescale assessments after a wildfire event. It is worth to be mentioned that some publications have analyzed the implementation of RUSLE after a fire event at different spatiotemporal scales [15–18].

Taking into consideration all the above, the major purpose and novelty of this work was to better understand and to explore synergistically the *NBR* and RUSLE methodologies before and after the wildfire event at three different periods of time. More specifically, the *NBR* index was calculated before and after the wildfire in order to delineate the boundaries of the total affected area. The following step was the implementation of the RUSLE parameters in the GIS software within the boundaries of the estimated area regarding the outputs from the *NBR* data sets. As a pilot area for the verification of the methodology's implementation, the affected area of Mazi, Malesina, was used. This is located in Central Greece, where extended areas of forest and agricultural land were burned. Furthermore, the soil loss rates that were calculated within the boundaries of burn severity zones mainly consist of agricultural regions combined with significant areas of natural vegetation. In addition, the high soil erosion areas derived from the applied methodology highlighted vulnerable areas in places where engineering works were constructed six years after the wildfire event.

## 2. Materials and Methods

### 2.1. Study Area

The study area (from 23°11′35″ E, 38°34′24″ N to 23°20′2″ E, 38°38′2″ N) (Figure 1) is located close to settlements Mazi, Malesina, and Martino, in the region of Central Greece and belongs to the Municipality of Lokroi. Its eastern part is surrounded by the North Euboean Gulf, while the burned area covers about 30 km$^2$.

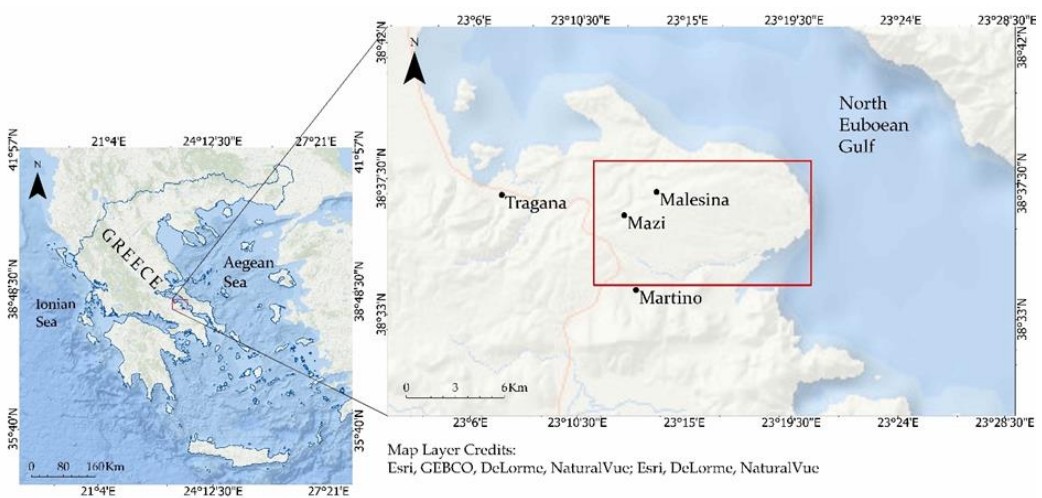

**Figure 1.** Location of the study area.

Regarding the geology of the study area, it is composed of Alpine rocks of the sub-pelagic zone (ophiolites) and post-Alpine formations. The ophiolites institutionalize the geological background and the ruptured ditches, which arose from the neotectonics activity of the faults and were covered by newer sediments of the Pleocene-Quaternary age. These sediments consist of sand, clay, and cohesive gravel [19]. The climate of the study area is Mediterranean, with heavy rainfall from October to March, while the prevailing winds are north and northwest. The driest months are July and August, while the months with the highest rainfall are December, January, and February [20]. The largest part of the wider region consists mostly of sclerophyllous vegetation, olive groves, and agricultural land, such as cultivation and natural grasslands. The examined area is characterized by varied topography combining different types of physical and geographical characteristics, such as the drainage network, which ends in the coastal area. As it is replicated in Figure 2, terrain elevation values do not exceed 450 m while the mean elevation is about 170 m. Vector hydrographic data for the study site were digitized from the Hellenic Military Geographical Service (HMGS) topographical sheet map 1:50,000 scale, including layers of streams, containing more than 30 features within the study area [21].

Within the boundaries of the burned area, the primary sector seems to dominate to a large extent, as the agricultural areas occupy quite large regions, such as complex farming systems, agricultural land, non-irrigable arable land, and olive trees. According to Figure 3, the major land cover consists of non-irrigable land (35%), nature conservation lands or national parks (30%), crown lands and reserves (16%), and others including intensive agriculture, horticulture, mining, and reservoirs (19%).

On 26 June 2014, a fire broke out in Mazi of Malesina, which burned more than 28 km$^2$ of agroforestry areas. Initially, the fire burned a forestry area in Mazi, and then due to adverse weather conditions, under the territory of strong winds and the existence of quite high temperatures, it spread to a pine-covered area. The fire was contained and brought under control on Sunday, 29 June, predominantly composed of trees established through regeneration, in the same location where another wildfire had occurred in the past.

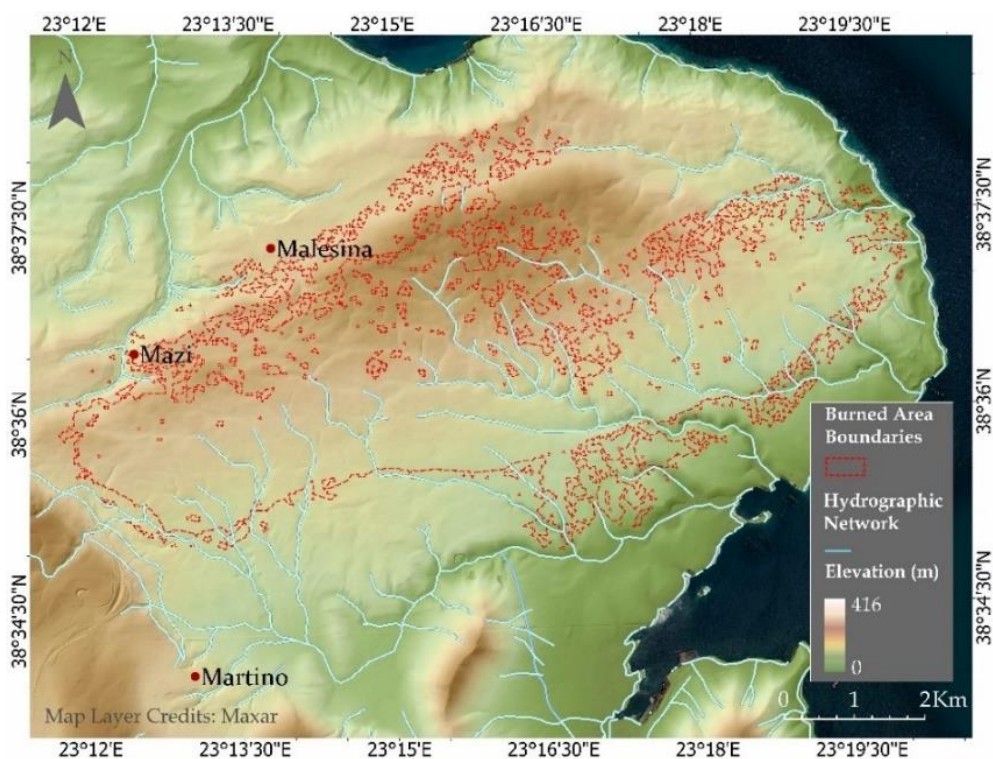

**Figure 2.** Geomorphological map of the study area. Red dashed polygon represents the boundaries of the affected area [22].

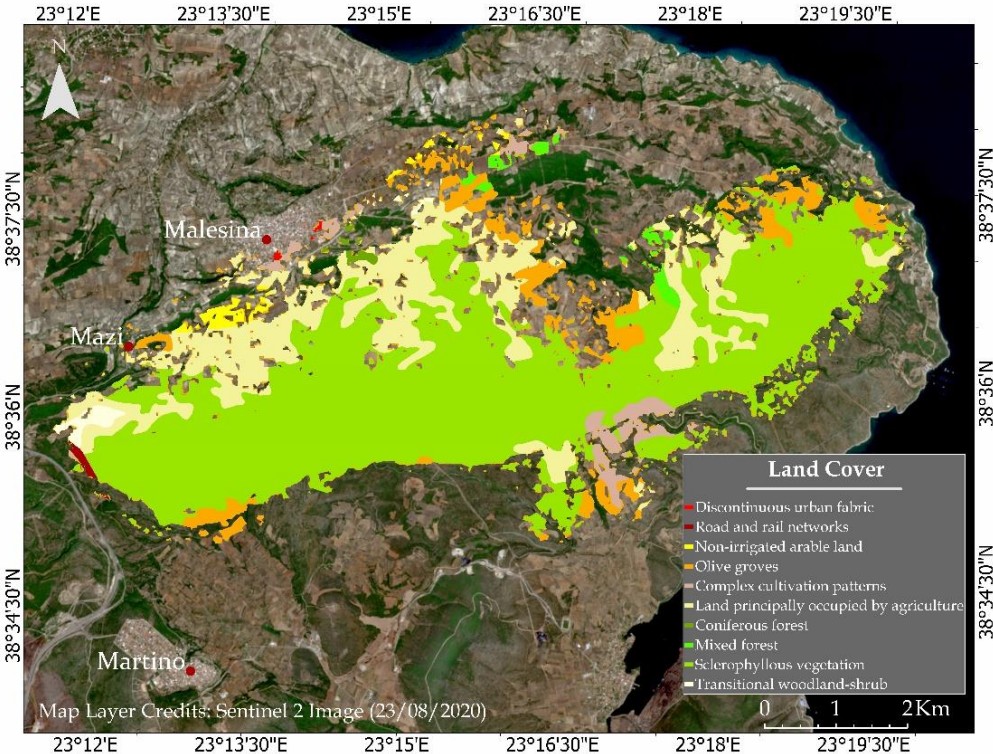

**Figure 3.** Land cover of the burned area according to Corine Land Cover 2012 [23].

### 2.2. Data

The first step was the delineation of the exact location related to the burned area (Figure 4) in order to assess the severity of the fire and to produce the final soil erosion products. For that purpose, Earth observation satellite images were acquired considering

the general footprint of the study area and the time of the wildfire event. Originally, the methodology of Miller and Thode [24] was used to accurately locate the burned area through the production of the burn severity index, where atmospherically corrected Landsat 8 images before and immediately after the fire were processed and analyzed. Furthermore, to produce the necessary factors of the RUSLE model, several data sets were acquired and analyzed. The implementation of this method was based on rainfall, soil, land cover, and topographical data. Rainfall data were obtained from five meteorological stations through the Meteo website of the Environmental Institute of the National Observatory of Athens (NOA) [20], covering a time before and after the fire, as well as more than five years after the wildfire event. In addition, some additional Earth observation data were obtained for soil loss assessment, acquired one year before the fire, one year after, and several years later, aiming to produce results regarding soil properties of the study area. Therefore, additional imagery was used, consisting of four Landsat 8 and two Sentinel-2A images, provided by the geoportal of United States Geological Survey (USGS) [25] and the European Union's "Copernicus" [26] program, respectively. Landsat 8 imagery was acquired in order to cover the time period of the wildfire event before the Sentinel-2 release date.

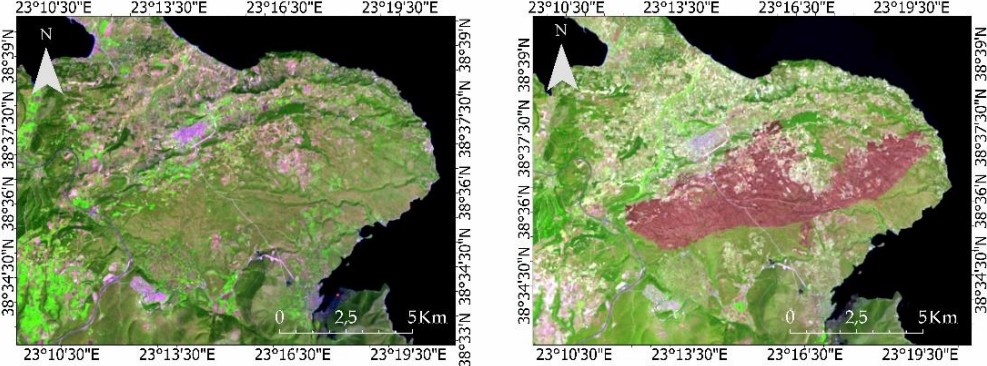

**Figure 4.** Infrared color Landsat 8 images before (**left**) and after (**right**) the fire event.

An important step was the information related to soil data, which was requested and provided by the European Soil Data Centre (ESDAC) of the Joint Research Centre (JRC) [27]. In addition, the topography was calculated from digital elevation model (DEM) with 5 m spatial resolution, which was derived from the Geospatial Data INSPIRE Geoportal of the "Hellenic Cadastre" [22]. The data were used to represent the topographical data of the RUSLE applied methodology. Moreover, to obtain the land use and/or land cover data of the study area, the Corine database was imported and processed for the years 2012 and 2018 [23]. It is worth to be mentioned that the land use and/or land cover data set was provided freely through the Copernicus website.

The following step was the integration of the homogenized and processed data into a GIS environment, where all the data sets were arranged in thematic layers, as presented in Table 1.

**Table 1.** Data sets used in the revised universal soil loss equation (RUSLE) model.

| Factors | Data Sets | Data Source | Spatial Scale | Temporal Scale | Primary Format |
|---|---|---|---|---|---|
| Rainfall erosivity (*R*) | Rainfall measurements by stations | National Observatory of Athens (NOA) | - | Daily, 2012–2020 | Vector (points) |
| Cover management (*C*) | Sentinel-2 and Landsat-8 images | Copernicus, USGS | 10 m (Sentinel-2A) 30 m (Landsat 8) | 2012–2013, 2014–2015, 2019–2020 | Raster (grid) |
| Soil erodibility (*K*) | Soil types | "European Soil Database-Soil Erodibility (*K*-Factor) High Resolution dataset for Europe" (ESDAC/JRC) | 500 m | 2014 | Vector (polygons) |
| Slope length and steepness (*LS*) | DEM | Hellenic Cadastre | 5 m | - | Raster (grid) |
| Support practice (*P*) | Land cover | Corine | 100 m | 2012, 2018 | Vector (polygons) |

*2.3. Burn Severity Index*

Burn Severity Index, as its name implies, is strongly related to fires regarding the spectral behavior of vegetation. Reflectance in the shortwave-infrared (*SWIR*), which is sensitive to soil and vegetation water content, increases after the fire, while in the near-infrared (*NIR*), there is a decrease in the reflectance due to the reduction in chlorophyll content of healthy vegetation [24,28]. In addition, the normalized burn ratio (*NBR*) has been developed to identify burned areas by combining near-infrared (*NIR*) and shortwave-infrared (*SWIR*) spectral areas [29,30]:

$$NBR = \frac{NIR - SWIR}{NIR + SWIR} \tag{1}$$

In this context, the *NBR* creation was based on Landsat 8 imagery in order to identify burned areas regarding the fire event in 2014 (Figure 5). Hence, the spectral bands 7 and 5 from Landsat 8 images were developed for the index implementation, corresponding to the spectral regions *NIR* and *SWIR*, respectively.

Subsequently, the differenced *NBR* (*dNBR*) provides an established and reliable tool for easier separation between burned and unburned vegetation, as well as for estimating the severity of the fire [31,32] and is calculated by removing the post-fire *NBR* image from the pre-fire *NBR* image [33]. The differenced index replicates the resulting comparison between the Landsat pre-fire *NBR* image and the post-fire *NBR* image so as to detect possible alterations since the fire event [34]. To perform a proper assessment of the *dNBR* index, the post-fire *NBR* image is required to have been obtained immediately after the cessation of fire activity [34] since the index is less effective if a period of time has elapsed during which vegetation regeneration processes have taken place.

To calculate the difference of the *NBR* index (*dNBR*), the following Equation is used:

$$dNBR = PrefireNBR - PostfireNBR \tag{2}$$

This index is used for the extraction of high accuracy burned area boundaries and the classification of specific categories so as to evaluate the severity of the fire. High *dNBR* values indicate high severity damage, while areas with negative *dNBR* values are more likely to indicate increased vegetation productivity after a fire incident, as mentioned in Table 2.

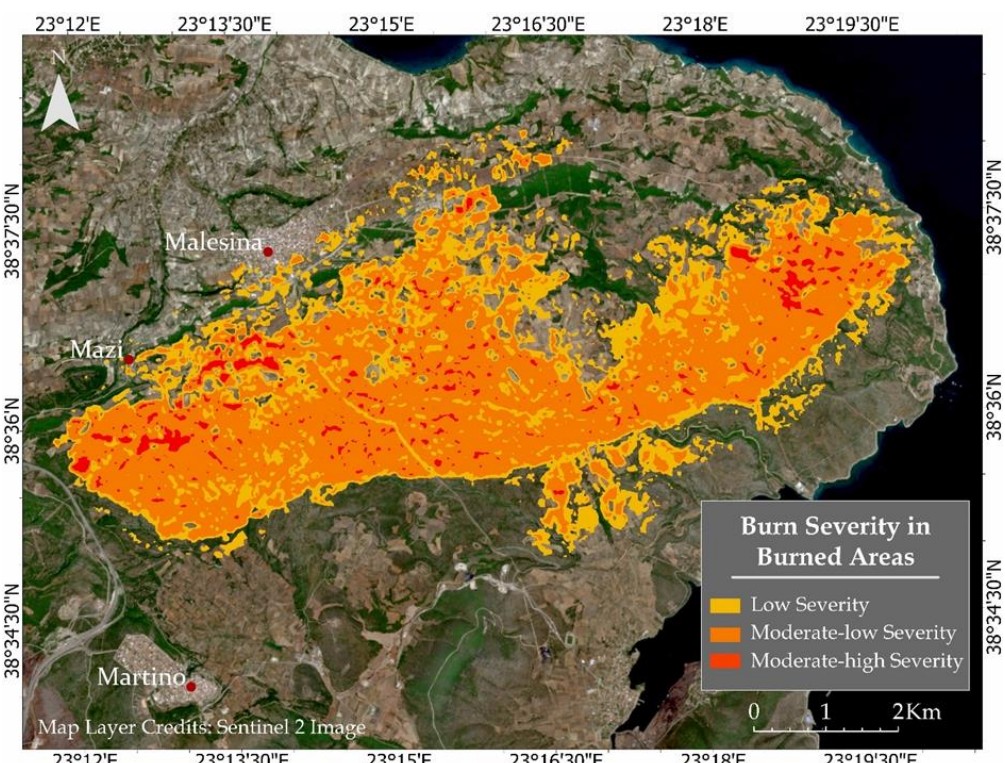

**Figure 5.** Burned areas classified according to burn severity of the study area.

**Table 2.** Burn severity based on *dNBR* values classified according to the USGS FireEffects Monitoring and Inventory Protocol (FireMON) program [35].

| Burn Severity | dNBR |
|:---:|:---:|
| Unburned | −0.1 to +0.09 |
| Low Severity | 0.1 to 0.26 |
| Moderate-Low Severity | 0.27 to 0.43 |
| Moderate-High Severity | 0.44 to 0.65 |
| High Severity | $\geq$0.66 |

### 2.4. Revised Universal Soil Loss Equation (RUSLE)

According to the literature, a wide variety of methodologies have been proposed for soil erosion assessment, with the most frequently used the RUSLE [6,36–38]. The RUSLE method [11] is a variant of the USLE method, first published in 1965 [39] for estimating average annual soil erosion values. During the evolution of satellites, data, and recent scientific results, the USLE method improved, and a revised version of this model (RUSLE) adopted integrating its ability to predict erosion by incorporating updated information made available through research over the last 40 years [11,40,41]. The revision and improvement of the USLE model factors created the RUSLE method, which was first published in 1991 [42]. The derived methodology was developed as an empirical equation for estimating soil loss caused by surface and rill erosion. Since then, RUSLE has been applied by a number of researchers at various scales and in a wide range of climates around the world [37,43–45].

More analytically, the RUSLE method is based on factors that determine soil erosion rates, according to the following Equation [11]:

$$A = R \times K \times LS \times C \times P \tag{3}$$

where *A* is the soil loss, *R* is the rainfall erosivity factor, *K* is the soil erodibility factor, *LS* is the slope length and steepness factor, *C* is the cover management factor, and *P* is the

support practice factor. These factors play a crucial role in defining soil erosion, while a miscalculation of any of them could lead to inaccurate assessment of the soil loss rates. Consequently, during the implementation of this method, much attention is needed so as to estimate the study area characteristics. Although RUSLE modeling is usually applied within the hydrological basin, in this research, the model was implemented within the burned area boundaries derived from *dNBR*. According to the units of the aforementioned factors in Equation (2), soil loss is calculated based on the soil erodibility (*K* factor) units for the time period defined by the rainfall erosivity (*R* factor) [43]. According to Gyssels et al. [46], soil loss decreases exponentially as vegetation increases. The rainfall erosivity factor *R* (MJ mm ha$^{-1}$ h$^{-1}$ season$^{-1}$) and the soil erodibility factor *K* (tn MJ$^{-1}$ mm$^{-1}$) have dimensions, while the rest of the model's factors are unitless. Their multiplication equals the total soil erosion in the study area.

However, by examining each factor one by one, the rainfall erosivity factor (*R* factor) consists of a key factor in evaluating the possibility of soil erosion development [40]. It is developed based on climatic data acquisition from meteorological stations and quantifies the effect of each rainfall episode [10]. Particularly, it defines the outcome of each rainfall episode's intensity and duration [47]. *R* factor is a fairly variable size on an annual, seasonal, and monthly basis. To estimate the *R* factor, climatic data were used, which were included in the linear correlation equation, originally developed for Portugal by Loureiro and Couthino [48], as well as for the Malesina region as follows:

$$R = \frac{\sum_{i=1}^{12}(7.05 \times r_{10} - 88.92 \times d_{10})}{N} \tag{4}$$

where $N$ is the number of months calculated annually, $r_{10}$ is the monthly rainfall exceeding 10 mm, and $d_{10}$ is the number of days with daily rainfall exceeding 10 mm per month. Thus, the *R* factor was estimated based on rainfall data extracted from each meteorological station on point data form representing each station. The resulting values were implemented on the spatial interpolation method, inverse distance weighting (IDW), through spatial analyst tool and interpolation toolset within the GIS environment.

*K* factor is the soil erodibility component that describes the sensitivity of soil particles to detachment and transport by rainfall and runoff [10] due to these processes and penetration [11]. In other words, the *K* factor represents the susceptibility to soil erosion, sediment transport, and the amount and rate of outflow [49]. The factor's value depends on the soil properties such as structure, organic matter, permeability, and texture-granulometry of the soil [50]. The estimation of erodibility factor was based on the ESDAC soil database, which includes the ready-made *K* factor for Europe, published by the JRC. *K* factor, which had a spatial resolution of 500 m per pixel, was converted into point data and then implemented on the spatial interpolation method Kriging.

The slope length and steepness factor (*LS* factor) signify the effect of topography on soil erosion [49]. This factor consists of two sub-factors, the slope length (L) and the slope steepness (S), whose multiplication determines the topographic *LS* factor. Both can be derived from DEM integrated into a GIS environment [51]. As stated by Panagos et al. [47], L is defined as the point of departure of the surface runoff to the point where either the slope decreases to such an extent that it is the starting point of the deposition process or the runoff focuses on a predetermined channel. *S* factor describes the behavior of soil erosion with an inclination angle. The approach developed by Moore and Burch [52] was applied to estimate the *LS* factor, where input data are distinguished in the upslope contributing area per unit width, determined by the flow accumulation, the pixel size, and the slope:

$$LS = \left(\frac{U}{L_0}\right)^m \times \left\{\left[\sin\left(\frac{\beta \times 0.01745}{S_0}\right)\right]^n\right\} \times (m+1), \tag{5}$$

where $U$ is the flow accumulation multiplied with the pixel size, $L_0$ is the slope length (22.13 m), $\beta$ is the slope in degrees, $S_0$ is the slope percentage (9%), $m$ is sheet erosion

ranging from 0.4 to 0.6, and *n* is rill erosion ranging from 1 to 1.3. The values of the indicators used were *m* = 0.6 and *n* = 1.1.

The root system enhances the mechanical properties (shear strength, slope stability, cohesion, etc.) of the soil, reduces the rate of surface runoff while maintaining and increasing filtration [53]. In RUSLE, the *C* factor evaluates the influence of soil cover, crop, and management territorial loss relative to territorial loss in bare fallow land areas [54]. It is a dimensionless factor ranging from 0 to 1, where values near 1 state the lack of coverage in vegetation and, therefore, the surface is considered to be bare, while values near 0 state intense coverage [55]. *C* factor was based on normalized difference vegetation index (NDVI) images, which were implemented in Equation (6). Specifically, four Landsat 8 products were used for the creation of seasonal maps before the fire incident, for the periods November 2012–February 2013 and May 2013–August 2013, and after the fire, November 2014–February 2015 and May 2015–August 2015, while also Sentinel 2 products were used examining a time period many years after the incident, on November 2019–February 2020 and May 2020–August 2020. Subsequently, the following formula was applied to these data according to Durigon et al. [56]:

$$C = \frac{1 - NDVI}{2} \qquad (6)$$

According to Shin [57], practices on agricultural land play a significant role in the land loss processes of the area. In obedience to Vidali [58], since these practices occupy a high rate of 36.57%, it was quite necessary to assess the impact of crop systems in soil erosion. *P* factor was created generated based on the Corine 2012 and 2018 land cover data. This factor was assigned values based on Yang et al. [59], where the CLC agricultural land categories (211, 212, 221, 222, 223, 231, 241, 242, 243) were given a value of 0.5, which according to David [60] corresponds to the "minimum plowing", while the rest of the classes were then given the value of 1. The support practice factor (*P* factor) is defined as the ratio of soil loss in a particular soil conservation practice compared to a field with elevation and soil elevation [13] and is a dimensionless size. Its value range is from about 0.2 to 1, with low *p* values corresponding to greater control of soil erosion. Due to the difficulty of capturing it, many studies ignore the assessment of the factor, giving it a value of 1.

The factors were set to GIS raster grids with 30 m of spatial resolution (pixel size). The majority of them were originally in raster format, while the rest were converted from vector to raster format. Therefore, the implementation of the final seasonal soil erosion maps of Mazi of Malesina for 2013, 2015, and 2020 (Figures 6 and 7) was achieved by incorporating in Equation (2), the invariant factors *K*, *LS*, and *P* and each of the seasonal factors, *R* and *C*.

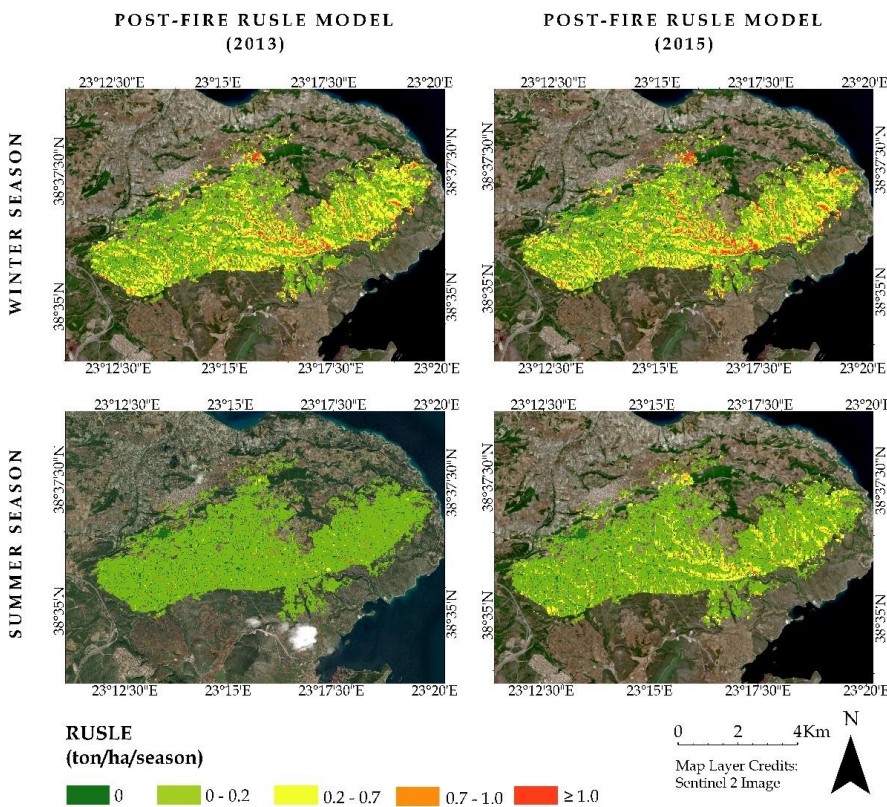

**Figure 6.** Seasonal spatial distribution of soil erosion rate for Mazi of Malesina in 2013 and 2015.

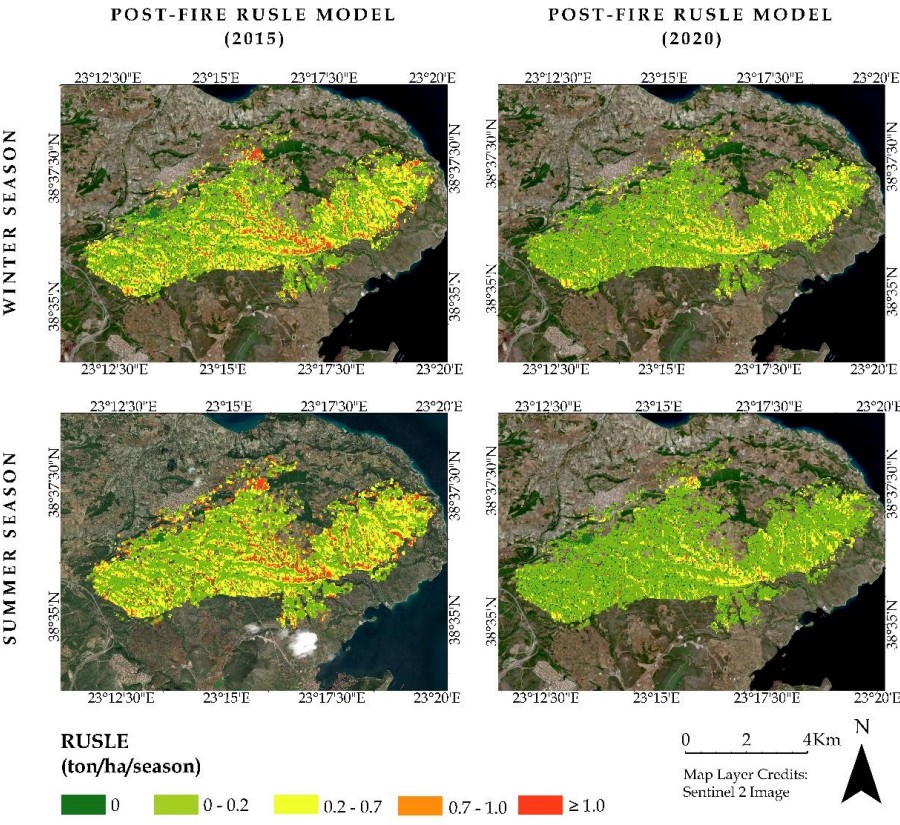

**Figure 7.** Seasonal spatial distribution of soil erosion rate for Mazi of Malesina in 2015 and 2020.

## 3. Results

### 3.1. Burn Severity Index Quantification

Figure 5 visualizes the spatial distribution of the evaluated Burn Severity Index for Mazi of Malesina after the fire case in June 2014. The study area displays a loss in high burn severity values, while the moderate-high severity-level accounts for 1.16 km², which corresponds to 4% of the total burned area and is consisted mostly of sclerophyllous vegetation (63.79%) and land principally occupied by agriculture with significant areas of natural vegetation (30.17%). Moderate-low severity values account for 18.15 km², with a percentage of 62.71% of the total burned area, which is the highest burn severity rate and is mainly characterized by sclerophyllous vegetation (74.9%) and land principally occupied by agriculture with significant areas of natural vegetation (17.91%) as well. Finally, the low severity level is attributed to the area of 9.64 km², 33.31% of the burned area mostly occupied by sclerophyllous vegetation (48.24%), and land principally occupied by agriculture with significant areas of natural vegetation (25.7%) as well.

### 3.2. Seasonal Spatial Variations of Soil Loss

Figures 6 and 7 represent the estimated soil loss, on a seasonal basis within the period of winter and summer, for Malesina in 2013, 2015, and 2020, while Figure 8 visualizes the soil loss values. It is observed that the seasonal maps of 2015 highlight the most erosive period after the wildfire event. Specifically, the winter seasonal map shows that the soil loss reaches up to 1.5 ton/ha/season exceeding the calculated soil erosion before the fire in 2013. In Figure 6, the difference is quite distinguishable between the pre-fire summer seasonal map and the post-fire one, while also Figure 9 indicates that the soil loss values have quadrupled regarding the initial ones before the fire incident.

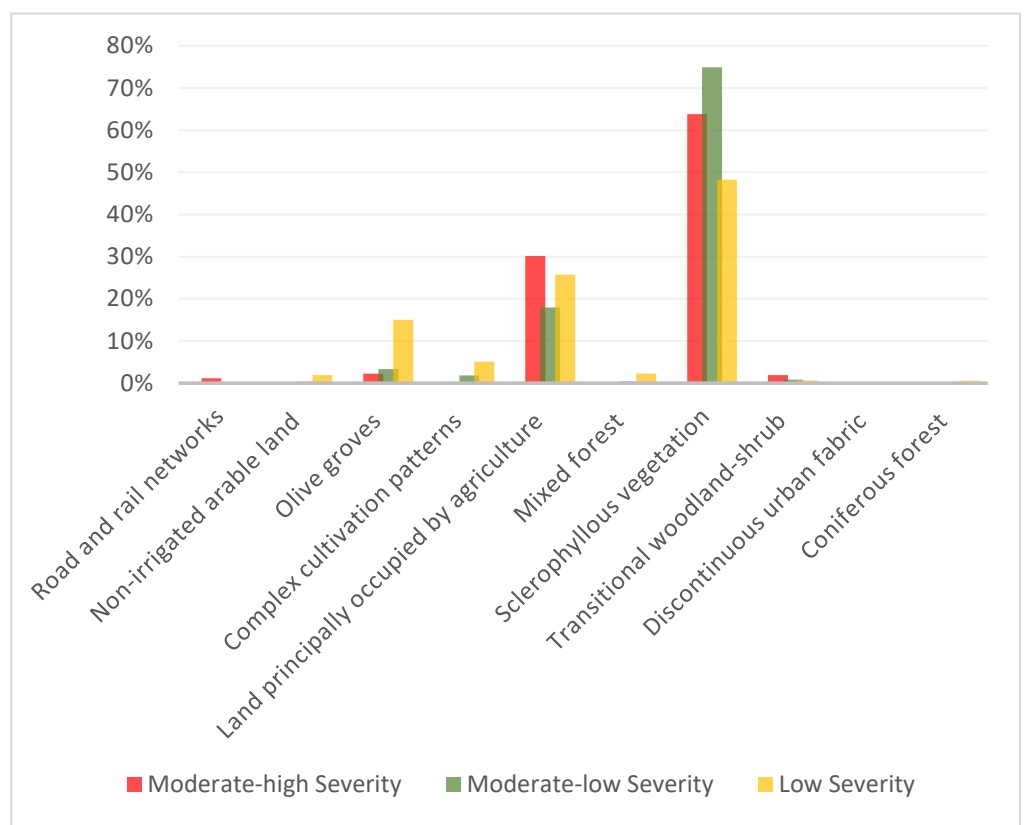

**Figure 8.** Burn severity quantification and land cover classification.

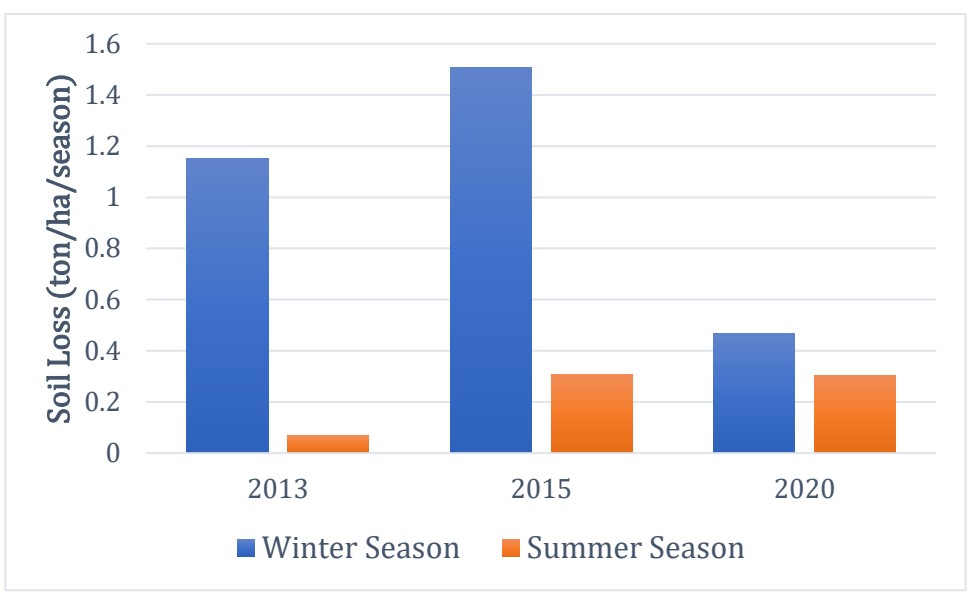

**Figure 9.** Seasonal variation of mean soil loss rate in 2013, 2015, and 2020.

On the contrary, during the period of 2020, soil loss significantly reduces, especially in winter, where the estimated number is lower than 0.47 ton/ha/season. Moreover, the corresponding summer seasonal maps have also been reduced, indicating minimal differences according to 2015. Soil loss values of 2020 have also reached values below those displayed in the corresponding seasonal maps before the fire incident in 2013.

Summarizing, regarding the summer periods, the derived maps represent low values of soil erosion with small fluctuations. Finally, results showed that there is a significant change between the pre-fire and post-fire seasonal soil loss values, while several years later, soil loss values seem to have decreased, reaching the minimum values of soil erosion.

### 3.3. Validation of the Results

According to the applied RUSLE approach, the highest erosion values occurred in winter 2015 (Figure 9). Thus, these spatial results could play a key role in an attempt to identify and locate the most vulnerable areas. More particularly, the model recognized areas adjacent to the drainage network that accumulate the highest soil erosion values.

In order to validate RUSLE modeling results, recent Google Earth images were analyzed and interpreted during 2019–2020. Combining optical observations and areas with high soil erosion rates, five areas were identified where infrastructure works were carried out, especially in the western part of the affected area (Figure 10). These areas validate RUSLE modeling results and emphasize the necessity for the development of engineering works in specific areas. More analytically, infrastructure works are recognized in Figure 9, located in areas where morphology is characterized by steep slopes due to the action of the drainage network, which run across cultivated lands on the plane region. Although a greater accumulation of soil loss values is located in the eastern part, where no engineering works have been carried out yet.

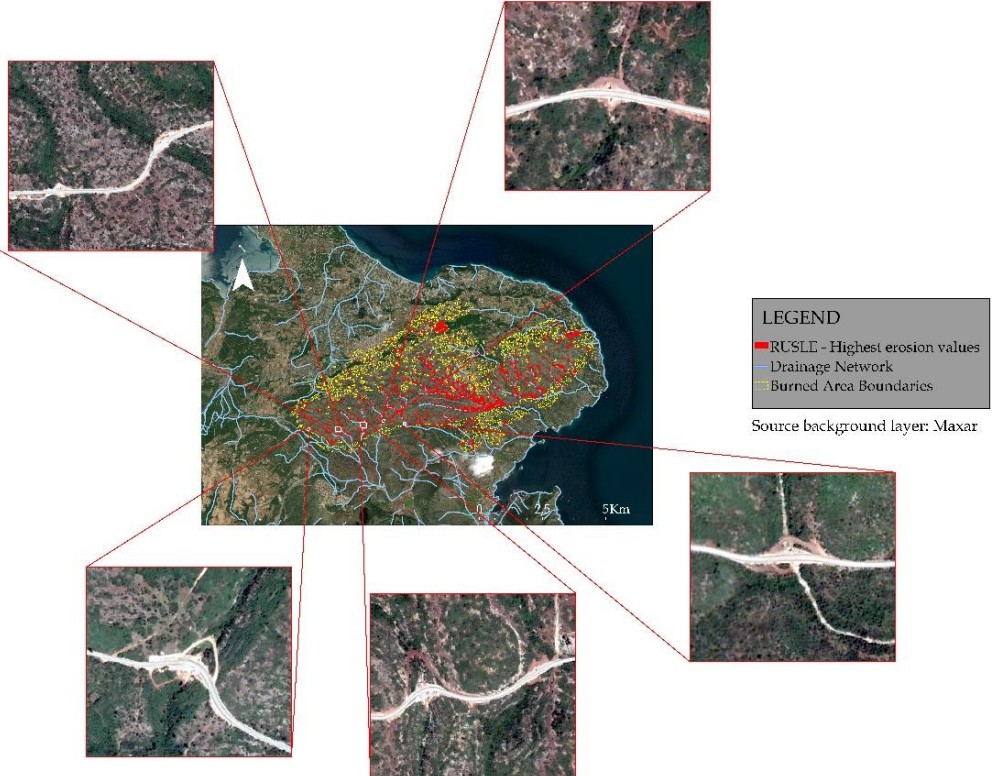

**Figure 10.** Google Earth imagery (obtained in 2021) shows depicts areas where engineering infrastructure works have been recently implemented. Yellow-shaped polygon represents the boundaries of the burned area.

## 4. Discussion

It is widely known that wildfires may result in extensive damages to properties, human losses and can destroy natural ecosystems [61]. Regarding European Union (EU), more than 85% of the total forest fires have occurred in the Mediterranean region [62]. Due to the nature of Mediterranean climate and other biological characteristics, forest fires and post-fire erosion take place every year [63]. Soil erosion is a very dynamic process affected by a wide variety of factors, such as the topographic position of slope, vegetation, and soil type, which play a crucial role in the behavior of soil erosion [53]. In compliance with that, this study estimated the seasonal variations of the aforementioned factors in order to assess the soil loss results for Malesina during the time periods 2013, 2015, and 2020. The implementation of RUSLE was mostly based on rainfall data and time-series satellite images, which formed the rainfall erosivity (*R*) and cover management (*C*) factors. The rest of the factors, which are the soil erodibility factor (*K*), the slope length and steepness factor (*LS*), and the support practices factor (*P*), were considered to be more "static" during the process of the creation of the seasonal soil loss results.

The RUSLE model is developed through parameters, each one of which is analyzed by the technology of geoinformation, providing a robust and cost-effective approach in examining their impact on soil erosion [36]. Despite the fact that RUSLE is identified as one of the most widely used soil erosion models, data availability factors are created, may lead to some limitations.

The RUSLE results were analyzed in order to calculate the soil erosion in the study area on a local scale (Figures 6 and 7). According to the occurring outputs, the seasonal maps of 2015 are identified as the most erosive period, which is after the fire break-out in 2014, while many years after the event, in 2020, soil erosion has significantly reduced as a result of the natural vegetation recovery. On the other hand, the calculation of recovery rates is still an issue that needs more research in terms of high resolution within the boundaries

of the Mediterranean region [64]. The outputs of RUSLE modeling regarding post-fire assessment seem to agree with those of other Mediterranean site researches [49] that distribute increased erosion rates after a fire incident. Regarding the spatial distribution, higher soil erosion values seem to be following the path of the hydrologic network. Areas near the drainage network seem to be characterized by steep slope lands leading to higher soil erosion activity as a result of increasing runoff velocity and erosion [10,65]. According to Bocchila et al. [66], drainage channel beds need to be cleaned systematically after a fire event due to the potential transportation of wooden debris or sediments to the hydrological basin to which the network belongs. Based on the seasonal maps, winter seasonal maps seem to represent maximum soil erosion values compared to the summer seasonal ones.

Under the prism of the examined factors, produced data sets are generated from different databases. Factors such as the cover management factor *C* and the rainfall erosivity factor *R*, which have been extracted from Sentinel-2 images and the National Observatory of Athens climate database, offer the advantage of comparing time-series data sets. However, the *C* factor represents a parameter that could be easily misinterpreted since some areas may be given values that correspond to burned land without necessarily having been burned. In addition, anthropogenic processes concerning the *P* factor play an important role in soil erosion control since it expresses the effect of cultivation processes on the reduction in soil erosion [43], ranging between values with a lack of corrosion control practices [10]. Thus, it is useful to include them in the erosion risk assessment, although there is no report that attributes global values of the factor, as the factor varies in relation to local activities [59].

Regarding the RUSLE method, some observations can be made to verify the reliability of this model. Initially, the databases from which the data were extracted play a significant role in their spatial resolution since high spatial resolution data sets contribute to the higher accuracy of the generated products. However, there were some limitations regarding the analysis of some factors, such as the support practice factor *P* and the soil erodibility factor *K*, that were derived from the Corine and JRC databases, respectively. These databases offer low spatial resolution products, leading to lower accuracy regarding the examination of the study area. On the contrary, the *LS* factor is produced from the DEM of the Hellenic Cadastre, which provides high spatial resolution at 5 m. Rainfall data, which was derived from the National Observatory of Athens database, was acquired from stations located far from the study area because of the lack of data at these time periods. Thus, the results' accuracy might be limited due to the restricted number of the available stations, reducing the *R* factor's reliability. Furthermore, factors *R* and *C* are considered to be determinants in defining the final outputs.

Subsequently, the applied methodology illustrated high-risk areas regarding soil erosion rates. Soil erosion hotspots, especially in the western part of the study area, were identified successfully. The implemented approach within the boundaries of the *dNBR* index identified five risk areas where engineering works occurred during 2019–2020. These observations are based on visual interpretation using timelapse images from Google Earth imagery (Figure 10).

According to the work of [67], RUSLE methodology does not take into consideration a few geomorphological processes such as gully, bank and channel erosion, and land movements. It is true that calculations of soil erosion rates are affected by different and complex parameters, and outputs from RUSLE modeling need ground-truthing. On the other hand, the RUSLE approach offers an important asset in visualizing the spatial distribution of soil erosion when fieldwork is abundant due to rough terrains and lack of resources. Finally, it is suggested that field surveys of soil erosion calculations are required in order to increase the accuracy and to better evaluate the reliability of outputs.

## 5. Conclusions

Taking into consideration the above mentioned, soil erosion is a global phenomenon that rises from agricultural intensification, terrestrial degradation, and manmade activities.

Consequently, its numerical assessment is very important due to the identification of anthropogenic and natural criteria. Modeling of soil erosion consists of a wide range of factors, which contribute to the creation of this phenomenon. Based on the above-mentioned findings, seasonal soil erosion maps visualized the spatial soil loss variations before and after the fire break-out that occurred in Malesina in 2014. Initial use of the *NBR* index and RUSLE model afterward within the boundaries of the affected area revealed some very interesting spatial patterns between the examined time periods in the study region since high erosion values are a consequence of the bare soils [68]. Specifically, the highest soil loss values were occurred in 2015, after the fire break-out in 2014, as a result of soil inefficiency in holding rainfall deposits. In addition, the study showed that regions vulnerable to high levels of soil erosion were identified to locations with direct tributaries to the major streams and steep sloping zones.

In the winter season of 2015, only a couple of months after the fire incident, a higher concentration of soil erosion values is observed, a fact that is verified according to Andreu et al. [69], who claim that post-fire regions seem to be more vulnerable to erosion 4 to 6 months after the fire incident. In addition, the study showed that regions vulnerable to high levels of soil erosion were identified to locations with direct tributaries to the major streams and steep sloping zones. Several years later, in 2020, the study area showed reduced soil erosion values indicating a remarkable regeneration reducing naturally the soil erosion rate, as predicted by the authors of [70], who claimed that soil loss values continuously decrease one year after the event. Moreover, winter seasonal maps were characterized by higher soil erosion values as opposed to the summer seasonal maps, which might be based on the fact that the *R* factor is higher in winter due to the rainfall quantity and frequency. Furthermore, in 2020 there seems to be a major change in the soil erosion of the area, mitigating the increased high values of 2015.

All in all, these findings could be essential for distinguishing the burned area effects on soil loss in Malesina as well as its spatial variations. From that point of view, the recognition of high soil loss time periods could add up to a strong basis for assessing the generation of potential measures. It is worth to be mentioned that the continuous implementation of RUSLE modeling can be useful as a monitoring soil erosion tool for areas with slope steepness and high altitudes. In addition, it contributes to the sustainable and logical decision making for the identification of vulnerable areas before and after a natural hazard. Furthermore, RUSLE replication can also indicate sensitive areas after fire incidents. Moreover, *dNBR* is a commonly used index in burned area delineation and burn severity classification, though in other Mediterranean regions, burn severity classes were driven by indexes such as LAI (Leaf Area Index) or NDVI [71]. To summarize, this study also underlines the significance of using remote sensing and geoinformation techniques to assess the post-fire effects on soil characteristics, not only on a regional but even more on a local scale.

Future work could be based on annual data sets of soil loss with higher spatial resolution in order to avoid limitations that may have some relation with the spatial distribution of the RUSLE model and Burn Severity Index. Further work could also include extending the knowledge of RUSLE parameters' associations to burn severity in post-fire erosion evaluation [49]. Additionally, as for the creation of some of its parameters, some data sets could be generated from different databases or satellite images in order to validate some of the key findings of this study. Field surveys using drone imagery would be an interesting approach that could overcome the difficulty of local-scale observations over a rolling landscape regarding the post-fire effects [71]. Finally, another potential for future work could be the creation or the implementation of another methodology in order to compare the current study's results.

**Author Contributions:** I.T. was involved in the computational framework, data analysis, results interpretation, and paper writing; P.K. was involved in order to implement geospatial techniques, interpret spatially derived data, and develop theoretical formalism; A.R. participated in conducting data interpretation and assisted in the paper writing; I.P. and N.K. had the general supervision of the study and provided solutions to any problems that arose. All authors have read and agreed to the published version of the manuscript.

**Funding:** This research received no external funding.

**Institutional Review Board Statement:** Not applicable.

**Informed Consent Statement:** Not applicable.

**Data Availability Statement:** Data available upon request.

**Acknowledgments:** Department of Geography, Harokopio University of Athens, provided the required facilities for this study, which the authors gratefully acknowledge. The authors are grateful to the European Space Agency and the National Aeronautics and Space Administration, who provided Sentinel-2 and Landsat 8 data accordingly. The authors would also like to thank the reviewers for providing useful suggestions that enhance the manuscript's quality.

**Conflicts of Interest:** The authors declare no conflict of interest.

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
