# Peer review of "Assessing Post-Fire Effects on Soil Loss Combining Burn Severity and Advanced Erosion Modeling in Malesina, Central Greece"

_remotesensing, doi:10.3390/rs13245160_

Round 1

Reviewer 1 Report

The article is interesting and discusses a very pertinent topic.

The structure of the manuscript is adequate and the methodology is also adequate and clearly explained. Sometimes is too much descriptive and should be synthetized.

The introduction is correct but the influence of wildfires in erosion should be deepen. Considering that the soil erosion methodology is specifically applied to burned areas, this fact should be adequately analyzed in introduction, even with some comparative values of erosion in burned and unburned areas.

The manuscript must be revised by an English native. There are several problems along the text (some are highlighted in the revised manuscript attached).

Figures are good, but I have some minor recommendations: in figure 2 the highest elevation values should be represented with darker colors; in figure 3 the legend is difficult to read.

Reviewer 2 Report

I have a conceptual question to ask. Regardless of the recurrence of forest fires, soil erosion is imminent in the winter seasons. Meanwhile, the RUSLE equation depends on several variables including the C, which is directly affected by the fires and upsurge erosion. I cannot find the synergy between soil erosion and the NBR in the article. What is your hypothesis?

Other comments;

It is not clear why the use if 2 optical sensors

114-117. unnecessary information

155-156. there is no need to mention any specific software; we are not supposed to promote commercial software’s

220-261. does not actually read as a method section, it is more like a discussion

Reviewer 3 Report

Dear authors, you describe an interesting topic from your region. 
The work is good, but needs improvement. It could gain value with that. Lately, mainly modeling is done. But models must also be validated.
Also the underlying calculation formulas must be included, even if they are familiar to the expert.
See the attached text for further comments.

Round 2

Reviewer 2 Report

Accept

Reviewer 3 Report

Dear authors, thank you for your response.

Please improve "assessed" in lit [20], I told you in v1
